



# Diurnal variations of BrONO₂ observed by MIPAS-B at mid-latitudes and in the Arctic

Gerald Wetzel[1], Hermann Oelhaf[1], Michael Höpfner[1], Felix Friedl-Vallon[1],
Andreas Ebersoldt[2], Thomas Gulde[1], Sebastian Kazarski[1], Oliver Kirner[3],
Anne Kleinert[1], Guido Maucher[1], Hans Nordmeyer[1], Johannes Orphal[1],
Roland Ruhnke[1], and Björn-Martin Sinnhuber[1]

[1]Karlsruhe Institute of Technology, Institute of Meteorology and Climate Research, Karlsruhe, Germany

[2]Karlsruhe Institute of Technology, Institute for Data Processing and Electronics, Karlsruhe, Germany

[3]Karlsruhe Institute of Technology, Steinbuch Centre for Computing, Karlsruhe, Germany

*Correspondence to:* Gerald Wetzel (gerald.wetzel@kit.edu)

## Abstract

The first stratospheric measurements of the diurnal variation of the inorganic bromine ($Br_y$) reservoir species $BrONO_2$ around sunrise and sunset are reported. Arctic flights of the balloon-borne Michelson Interferometer for Passive Atmospheric Sounding (MIPAS-B) were carried out from Kiruna (68°N, Sweden) in January 2010 and March 2011 inside the stratospheric polar vortices where diurnal variations of $BrONO_2$ around sunrise have been observed. High nighttime $BrONO_2$ volume mixing ratios of up to 21 parts per trillion by volume (pptv) were detected in the late winter 2011 in the absence of polar stratospheric clouds (PSCs). In contrast, the amount of measured $BrONO_2$ was significantly lower in January 2010 due to low available $NO_2$ amounts (for the build-up of $BrONO_2$), heterogeneous destruction of $BrONO_2$ on PSC particles, and the gas-phase interaction of BrO (the source to form $BrONO_2$) with ClO. A further balloon flight took place at mid-latitudes from Timmins (49°N, Canada) in September 2014. Mean $BrONO_2$ mixing ratios of 22 pptv were observed after sunset in the altitude region between 21 and 29 km. Measurements are compared and discussed with the results of a multi-year simulation performed with the chemistry climate model ECHAM5/MESSy Atmospheric Chemistry (EMAC). The calculated temporal variation of $BrONO_2$ is in principal agreement with the balloon-borne observations. Using the nighttime simulated ratio between $BrONO_2$ and $Br_y$, the amount of $Br_y$ observed by MIPAS-B was estimated to about 21-25 pptv in the lower stratosphere.





## 1 Introduction

Chlorine and bromine species play a dominant role in the contribution to ongoing stratospheric ozone depletion since the amount of equivalent effective stratospheric chlorine (chlorine and bromine) is predicted to return to 1980 values by 2050 at mid-latitudes (Newman et al., 2007; Stolarski et al., 2010). $BrONO_2$ is, besides BrO, the most abundant inorganic bromine ($Br_y$) compound in the stratosphere (see e.g. Brasseur and Solomon, 2005; Sinnhuber et al., 2009; Sinnhuber and Meul, 2015). $BrONO_2$ is formed via the reaction with BrO and $NO_2$:

$$BrO + NO_2 + M \rightarrow BrONO_2 + M. \tag{R1}$$

During day, $BrONO_2$ is photolyzed with different possible channels

$$BrONO_2 + h\nu \rightarrow Br + NO_3 \tag{R2a}$$
$$\rightarrow BrO + NO_2 \tag{R2b}$$

with a higher quantum yield of (R2a) compared to (R2b). $BrONO_2$ can also be destroyed via the reaction with atomic oxygen:

$$BrONO_2 + O(^3P) \rightarrow BrO + NO_3. \tag{R3}$$

(R1) to (R3) exhibit the close connection between BrO and $BrONO_2$ leading to an opposite diurnal variation of these species. Gas-phase $BrONO_2$ can also be converted to gas-phase HOBr and BrCl on sulphate aerosols and polar stratospheric cloud (PSC) particles where $H_2O$, HCl, and $HNO_3$ are in liquid (l) or solid phase (s):

$$BrONO_2 + H_2O \,(l, s) \rightarrow HOBr + HNO_3 \,(l, s) \tag{R4}$$

$$BrONO_2 + HCl \,(l, s) \rightarrow BrCl + HNO_3 \,(l, s). \tag{R5}$$

An interaction between the chlorine and bromine family (particularly important at high latitudes in winter under conditions of elevated ClO) is the gas-phase production of BrCl via:

$$ClO + BrO \rightarrow BrCl + O_2. \tag{R6}$$

Stratospheric $BrONO_2$ was detected for the first time by the Michelson Interferometer for Passive Atmospheric Sounding (MIPAS) aboard the Envisat satellite (Höpfner et al., 2009). Strong day/night variations were observed with much lower concentrations during day compared to nighttime. A maximum amount of 20-25 pptv (parts per trillion by volume) was inferred from MIPAS spectra recorded during the night.





Flights of the balloon version of the MIPAS instrument (MIPAS-B) investigated in this work
were carried out from Kiruna (68°N, Sweden) on 24 January 2010 and 31 March 2011 as well
as from Timmins (49°N, Canada), on 7/8 September 2014. For the first time, diurnal variations
of $BrONO_2$ around sunrise (Kiruna flights) and sunset (Timmins flight) were measured by
MIPAS-B with high temporal resolution. A description of the MIPAS-B instrument, data
analysis and chemical modelling is given in Sect. 2. A discussion of observed $BrONO_2$ volume
mixing ratio (VMR) vertical profiles follows in Sect. 3 together with a comparison of the
measured data to simulations of the chemistry climate model ECHAM5/MESSy Atmospheric
Chemistry (EMAC) to check the current understanding of stratospheric bromine chemistry and
to estimate the amount of lower stratospheric $Br_y$.

## 2    MIPAS-B instrument, data analysis and modelling

In the following sections, we give an overview of the MIPAS-B instrument and the balloon
flights together with the corresponding data analysis and a description of chemical modelling
performed for this study.

### 2.1    MIPAS-B instrument and balloon flights

The balloon-borne cryogenic Fourier Transform limb emission spectrometer operates in the
mid-infrared spectral region between about 4 and 14 µm. The maximum optical path difference
of 14.5 cm of the beam in the interferometer correlates with 0.0345 $cm^{-1}$ spectral resolution.
This corresponds to about 0.07 $cm^{-1}$ after apodization with the Norton and Beer (1976) "strong"
function and allows the separation of individual spectral lines from continuum-like emissions.
Noise equivalent spectral radiance (NESR) values for a single calibrated spectrum are typically
within $1 \times 10^{-9}$ and $7 \times 10^{-9}$ W($cm^2$ sr $cm^{-1}$)$^{-1}$. A reduction of spectral noise by a factor of $n^{-0.5}$ is
obtained by recording and averaging of $n$ spectra ($n \leq 16$) per single elevation scan. Besides a
high radiometric accuracy of typically 1%, the pointing system allows a knowledge of the
tangent altitude of better than 50 m at the 1-σ confidence limit. An overview of instrument
characterization in terms of the instrumental line shape, field of view, NESR, line of sight of
the instrument, detector non-linearity (Kleinert, 2006) and the error assessment of the calibrated
spectra is given by Friedl-Vallon et al. (2004).

In this study, we report $BrONO_2$ results from three MIPAS-B flights. Details are shown in Table
1. The first flight took place on 24 January 2010 from Kiruna over northern Scandinavia inside





the Arctic vortex at the beginning of a major stratospheric warming (Wetzel et al., 2012). The second one was carried out from the same location on 31 March 2011 inside a still persistent late-winter Arctic vortex (Wetzel et al., 2015). The third one was performed at mid-latitudes from Timmins (Ontario, Canada) on 7 to 8 September 2014. For this mid-latitude flight, we show retrieval results from spectra observed around sunset. For the Arctic flights, MIPAS-B

measurements were performed from night into day. All flights have in common that fast sequences of spectra were recorded in short time steps of about 10 min to enable the retrieval of photochemically active species, which change quickly their concentration around sunrise and sunset. The line of sight of the instrument was aligned perpendicular to the azimuth direction of the sun to allow for a symmetric illumination of the sounded air mass before and beyond the

tangent point. The analysis of the recorded spectra is described in the following section.

## 2.2 Data analysis

Radiance calculations were carried out with the Karlsruhe Optimized and Precise Radiative transfer Algorithm (KOPRA; Stiller et al., 2002). Spectroscopic parameters for the calculation of emission spectra were taken from the high-resolution transmission molecular absorption

database (HITRAN; Rothman et al., 2009) and a MIPAS dedicated line list (Raspollini et al., 2013). Spectral features of the molecule $BrONO_2$ were calculated using new pressure-temperature dependent absorption cross sections measured by Wagner and Birk (2016) with a 2% intensity accuracy. KOPRA also provides derivatives of the radiance spectrum with respect to atmospheric state and instrument parameters (Jacobians) which are used by the retrieval

procedure KOPRAFIT (Höpfner et al., 2002). The vertical distance of tangent altitudes ranges between 1 and 1.5 km. Thus, the retrieval grid was set to 1 km up to the balloon float (observer) altitude. Above this level, the vertical spacing increases gradually up to 10 km at the top altitude of 100 km. Considering the smoothing of the vertical part of the instrumental field of view, the retrieval grid is somewhat finer than the achievable vertical resolution of the measurement for

most parts of the altitude region covered (especially above the observer altitude). To avoid retrieval instabilities caused by this oversampling, a Tikhonov-Phillips regularization approach (Phillips, 1962; Tikhonov, 1963) was applied using a constraint with respect to a first derivative of the a priori profile $x_a$ of the target species:

$$x_{i+1} = x_i + [K_i^T S_y^{-1} K_i + R]^{-1} [K_i^T S_y^{-1}(y_{meas} - y(x_i)) - R(x_i - x_a)],$$ (1)


where $x_{i+1}$ is the vector of the state parameters $x_i$ for iteration $i+1$; $y_{meas}$ is the measured radiance

vector and $y(x_i)$ the calculated radiance using state parameters of iteration $i$; $\mathbf{K}$ is the Jacobian

matrix with partial derivatives $\partial y(x_i)/\partial x_i$ while $\mathbf{S}_y^{-1}$ is the inverse noise measurement covariance

matrix and $\mathbf{R}$ a regularization matrix composed of the first derivative operator and a

regularization strength parameter.

The $BrONO_2$ retrieval calculations were performed in the range of the $\nu_3$ band centred at 803.37

$cm^{-1}$. Figure 1 shows spectral contributions of relevant species in the $BrONO_2$ microwindow

from 801 to 820 $cm^{-1}$ that has been found best appropriate to derive the $BrONO_2$ amount from

MIPAS-B spectra. Besides the target molecule $BrONO_2$, all main interfering species $H_2O$, $CO_2$,

$O_3$, $NO_2$, $HNO_3$, $COF_2$, HCFC-22 ($CHClF_2$), $CCl_4$, CFC-113 ($C_2Cl_3F_3$), $ClONO_2$, $HO_2NO_2$, and

PAN (peroxyacetyl nitrate) were fitted simultaneously together with temperature, instrumental

offset and wavenumber shift. The molecule $HO_2NO_2$ shows a similar spectral band shape like

the target species $BrONO_2$. Since the $HO_2NO_2$ absorption cross sections (included in HITRAN)

measured by May and Friedl (1993) are derived at only one temperature (220 K) a second set

of cross sections derived by Friedl et al. (1994) at room temperature (298 K) was used to allow

a two-point interpolation of the cross section intensity to the current atmospheric temperature.

Vertical profiles of minor contributing species were either adjusted in appropriate

microwindows prior to the $BrONO_2$ retrieval or taken from a climatological atmosphere

(Remedios et al., 2007), updated with surface concentration data from NOAA ESRL GMD

(National Oceanic and Atmospheric Administration, Earth System Research Laboratory,

Global Monitoring Division; Montzka et al., 1999). An example of a best fit of a measured

MIPAS-B spectrum zoomed around the Q-branch region of the $BrONO_2$ $\nu_3$ band for a tangent

altitude near 20 km is shown in Figure 2. The spectrum was recorded during night. If the fit is

performed in absence of $BrONO_2$ in the model atmosphere, a systematic residual is remaining

around the centre of the $BrONO_2$ Q-branch at 803.37 $cm^{-1}$ (blue solid line in Figure 2). If the

molecule $BrONO_2$ is taken into account by the radiative transfer calculation, the systematic

residual around the Q-branch disappears demonstrating the existence of $BrONO_2$ in the

stratosphere. Another example for a best fit in the same altitude region but for a MIPAS-B

spectrum recorded during day is illustrated in Figure 3. Here, we recognize that for a daytime

situation the effect whether the species $BrONO_2$ is included in the radiative transfer calculations

or not, is clearly smaller compared to the nighttime case (cf. Figure 2) such that we expect lower

stratospheric $BrONO_2$ VMRs during day and higher values at night. This is confirmed by the





retrieved vertical profiles of $BrONO_2$ illustrated in Figures 4 and 5 together with the error budget and altitude resolution. The dominant part of the total error in the $BrONO_2$ retrieval is spectral (random) noise resulting in a $BrONO_2$ VMR error of about 2 to 4 pptv (10-25%) in the altitude region of the VMR maximum. An important systematic error source are uncertainties of disturbing gases overlapping the $BrONO_2$ $\nu_3$ band. This influence was estimated using uncertainties in line intensity and half-width as given by Flaud et al. (2003) and HITRAN (Rothman et al., 2009) and results into a $BrONO_2$ error of up to 2 pptv (10-20%) in the altitude region of the $BrONO_2$ VMR maximum. Retrieval simulations of the major interfering species $O_3$, $CO_2$, and $H_2O$ have revealed an influence (line half-width and intensity uncertainties) within 10% on the $BrONO_2$ amount (Höpfner et al., 2009). The species $ClONO_2$, followed by $HO_2NO_2$ have large contributions to the limb emission spectra (see Figure 1). Temperature and pressure dependent $ClONO_2$ absorption cross sections were measured by Wagner and Birk (2003) with high accuracy. Systematic errors in the $BrONO_2$ VMR due to $ClONO_2$ spectroscopy are expected to be within 10% (Wagner and Birk, 2016). As mentioned above, a temperature dependence of $HO_2NO_2$ absorption cross sections was included to improve spectroscopy of this interfering species. Further systematic error sources like radiometric gain, line of sight, and the spectroscopy of the target molecule $BrONO_2$ itself are of minor importance for the total error budget of the $BrONO_2$ retrieval (see Figures 4 and 5). The altitude resolution of the retrieved $BrONO_2$ profiles amounts to between about 4 and 6 km (~ 4-5 degrees of freedom) over a wide range in the stratosphere.

## 2.3 Model calculations

Measured MIPAS-B data are compared to a multi-year simulation of the chemistry climate model EMAC that includes sub-models describing tropospheric and middle atmosphere processes (Jöckel et al., 2010). The core atmospheric model is the 5th generation European Centre Hamburg general circulation model (ECHAM5; Roeckner et al., 2006) which is linked to the sub-models via the interface Modular Earth Submodel System (MESSy). For the present study we applied EMAC (ECHAM5 version 5.3.02, MESSy version 2.52) in the T42L90MA-resolution, i.e., with a spherical truncation of T42 (corresponding to a Gaussian grid of approximately 2.8 by 2.8 degrees in latitude and longitude) and 90 vertical hybrid pressure levels from the ground up to 0.01 hPa (approx. 80 km). The calculation of gas-phase chemistry is realized by the submodel MECCA (Sander et al., 2005). The submodel MSBM (Kirner et al., 2011) simulates polar stratospheric clouds and calculates heterogeneous reaction rates.





A Newtonian relaxation technique of the surface pressure and the prognostic variables
temperature, vorticity, and divergence above the boundary layer and below 1 hPa towards the
ECMWF reanalysis ERA-Interim (Dee et al., 2011) has been applied to simulate realistic
synoptic conditions (van Aalst, 2005). The simulation includes a comprehensive chemistry
setup from the troposphere to the lower mesosphere with more than 100 species involved in gas
phase-, photolysis-, and heterogeneous reactions on liquid sulphate aerosols, nitric acid
trihydrate (NAT) and ice particles. Rate constants of gas-phase reactions originate from
Atkinson et al. (2007) and the Jet Propulsion Laboratory (JPL) compilation (Sander et al.,
2011). Photochemical reactions of short-lived bromine-containing organic compounds $CH_3Br$,
$CHBr_3$, $CH_2Br_2$, $CH_2ClBr$, $CHClBr_2$, and $CHCl_2Br$ are integrated into the model setup (Jöckel
et al., 2016). Surface emissions of these species are taken from scenario 5 of Warwick et al.
(2006). During the time period with MIPAS-B balloon flights the model output data were saved
every 10 minutes. The temporally closest model output to the MIPAS-B measurements was
interpolated in space to the observed geolocations.

## 3    Results and discussion

In this section, vertical profiles retrieved from MIPAS-B limb emission spectra measured
before and after sunrise (Arctic flights) and sunset (mid-latitude flight) are shown. The
measured data have been temporally smoothed with a 3-point adjacent averaging routine to
attenuate noisy structures. These data were compared to EMAC simulations. To permit a more
realistic comparison with respect to different altitude resolutions in the measurement and the
simulation, EMAC vertical profiles were additionally smoothed with the averaging kernel
matrix and the a priori profile of MIPAS-B. A smoothed EMAC profile $x_s$ is calculated
following the method described in Rodgers (2000):

$$x_s = x_a + \mathbf{A}(x - x_a^*),$$    (2)

where $x_a$ is the a priori profile of MIPAS-B, $x_a^*$ the a priori profile interpolated to the altitude
grid of the EMAC profile $x$, and $\mathbf{A}$ is the averaging kernel matrix of MIPAS-B.

### 3.1    Arctic measurements

The temporal evolution of $BrONO_2$ measured during the balloon flight from Kiruna on 31
March 2011 inside the late winter stratospheric polar vortex is shown in Figure 6. No PSCs





were present during the time of the MIPAS-B measurement (Wetzel et al., 2015). A nighttime

maximum of $BrONO_2$ around 25 km with values of more than 20 pptv is clearly visible. After
sunrise, the amount of $BrONO_2$ decreases to maximum values of about 14 pptv around 22 km.
The corresponding EMAC simulation is depicted in Figure 7. The overall structure of the
simulated temporal evolution of $BrONO_2$ is similar to the measured one. Maximum nighttime
$BrONO_2$ values in EMAC are comparable to the measured amounts. However, above the

nocturnal VMR maximum, EMAC calculates higher $BrONO_2$ concentrations compared to the
balloon observation. Furthermore, the daytime photochemical destruction of $BrONO_2$ is
slightly faster in the model yielding several pptv lower daytime $BrONO_2$ VMRs in the model
compared to MIPAS-B. A sensitivity study performed by Kreycy et al. (2013) indicates that
very likely the $BrONO_2$ photolysis rate and the reaction rate of the $BrONO_2$ build-up from BrO

and $NO_2$ differs from the JPL recommendation that was also used in EMAC. However, their
study refers to stratospheric mid-latitude conditions and the outcome is therefore not directly
comparable to the Arctic observations shown here. The EMAC simulation smoothed with the
averaging kernel matrix of MIPAS-B according to Eq. (2) is displayed in Figure 8. A main
difference to the unsmoothed case shown in Figure 7 is the reduction of the nighttime $BrONO_2$

VMR at altitudes above the maximum that yields to a better agreement with measured $BrONO_2$.

During night, a large part of lower stratospheric inorganic bromine is in the form of $BrONO_2$.
This gives the opportunity to estimate the amount of "measured" inorganic bromine $[Br_y(meas)]$
from measured nighttime $[BrONO_2(meas)]$ using the calculated $[BrONO_2(mod)]/[Br_y(mod)]$
ratio from EMAC in the following form:

$$[Br_y(meas)] = \frac{[BrONO_2(meas)][Br_y(mod)]}{[BrONO_2(mod)]}. \tag{3}$$

Applying Eq. (3) for a nighttime (SZA $\geq$ 96°) ratio $[BrONO_2(mod)]/[Br_y(mod)] \geq 0.8$
corresponding to an altitude region between 23 and 29 km we calculate $[Br_y(meas)]$ (including
the total $[BrONO_2(meas)]$ error) to $22.3 \pm 2.2$ pptv. The given error bar represents the 1-σ total

error originating from measured $BrONO_2$.

Another Arctic balloon flight was performed from Kiruna on 24 January 2010 inside a cold
polar vortex under mid-winter weak illumination conditions. As a consequence of low
stratospheric temperatures in this winter, widespread PSCs were present in an altitude region
between about 18 and 24 km at the time of the MIPAS-B observation (Wetzel et al., 2012). The





observed BrONO$_2$ as seen from night until noon is shown in Figure 9. Nighttime BrONO$_2$ mixing ratios are clearly lower compared to the previously discussed situation in late March 2011. This is also reflected in the EMAC simulation (Figure 10). During the long polar night the amount of available NO$_2$ (Wetzel et al. 2012) to produce BrONO$_2$ via (R1) is significantly reduced due to the conversion of NO$_2$ into its reservoir species (mainly HNO$_3$). In this period

of darkness, nearly all BrONO$_2$ below 25 km (PSC region) is converted to BrCl via heterogeneous chemistry according to (R5) and gas-phase conversion of BrO to BrCl via (R6). Here, more than 90 % of Br$_y$ are in the form of BrCl in the model simulation during night. Above this altitude region, BrONO$_2$ and BrCl together are the dominant species of the nocturnal Br$_y$ budget in the EMAC run. During day, photolyzation of both species (BrONO$_2$ and BrCl)

leads to an increase of BrO such that this species then dominates the Br$_y$ budget. If we smooth the EMAC BrONO$_2$ data with the averaging kernel matrix of MIPAS-B, we see a better agreement with MIPAS-B in the structure of the temporal evolution of the BrONO$_2$ amount (Figure 11). The effect of the smoothing appears to be stronger compared to the case in March 2011 since low temperatures together with low amounts of BrONO$_2$ in January 2010 entailed

to perform the retrieval with a factor of 2 coarser altitude resolution compared to a standard BrONO$_2$ retrieval setup as depicted in Figures 4 and 5.

### 3.2    Mid-latitude measurements

MIPAS-B spectra have been recorded from day until night over Ontario (Canada) during a balloon flight launched from Timmins on 7 September 2014. The temporal evolution of

measured BrONO$_2$ is depicted in Figure 12. A significant increase of BrONO$_2$ starting shortly before sunset is visible. This is caused by the weakened illumination at SZAs near 90° that enables the build-up of BrONO$_2$ from daytime BrO via (R1). Nighttime BrONO$_2$ mixing ratios of more than 24 pptv are seen by MIPAS-B around 28 km altitude. The corresponding EMAC model simulation is displayed in Figure 13. The principal shape of the increase of BrONO$_2$

VMR is reproduced by the model run although absolute values are somewhat lower in the simulation compared to the measurement. Differences in absolute BrONO$_2$ amounts are at least partly connected with the fact that EMAC NO$_2$ values are up to 20 % lower than the observed NO$_2$ in the altitude region of the BrONO$_2$ VMR maximum. The BrONO$_2$ increase in the model starts earlier compared to the measurement. Nighttime maximum BrONO$_2$ values in EMAC

reach about 22 pptv and are located in the same altitude region as seen in the observation. Smoothing the EMAC data with the averaging kernel matrix of MIPAS-B yields to a better



agreement with the structure of the observational data at altitudes below about 18 km (Figure 14).

During night, more than 90 % of simulated mid-latitudinal lower stratospheric inorganic
bromine is in the form of $BrONO_2$. Hence, we again apply Eq. (3) to estimate "measured" inorganic bromine. In an altitude region between 21 and 29 km, corresponding to a nighttime (SZA ≥ 99°) ratio $[BrONO_2(mod)]/[Br_y(mod)]$ ≥ 0.9, we then calculate $[Br_y(meas)]$ to 23.6 ± 1.9 pptv.

**4    Conclusions**

$BrONO_2$ observations around sunrise were performed during balloon flights with MIPAS-B carried out in the Arctic from Kiruna on 24 January 2010 and 31 March 2011 and at mid-latitudes from Timmins on 7/8 September 2014. Measured $BrONO_2$ diurnal variations with high nightime and low daytime values confirm the stratospheric bromine chemistry (introduced
in Sect. 1) that is dominated by the interaction of BrO and $BrONO_2$ according to (R1) – (R3). During polar winter (January 2010) with weak illumination, large parts of nighttime $Br_y$ are in the form of BrCl resulting in significantly lower $BrONO_2$ values compared to the situation in late Arctic winter (March 2011) and mid-latitude summer (September 2014).

The chemistry climate model EMAC is able to reproduce the temporal variation of the measured
$BrONO_2$ values. However, some differences in the absolute amounts of $BrONO_2$ are obvious. The simulated $BrONO_2$ mixing ratios are dependent on the assumed total $Br_y$ in the model, which amounts about 23 pptv in the lower stratosphere. As mentioned in Sect. 2.3 reactions of short-lived bromine-containing organic compounds are integrated into the model setup according to emission scenarios shown by Warwick et al. (2006). This is equivalent to about 6-
7 pptv inorganic bromine from these oceanic short-lived bromocarbons in the upper troposphere.

As discussed in Sect. 3, $Br_y$ in the lower stratosphere was estimated from MIPAS-B measurements. For the Arctic observation in March 2011, we obtain 22.3 ± 2.2 pptv $Br_y$ and for the mid-latitude measurement in September 2014, we calculate 23.6 ± 1.9 pptv $Br_y$ in the
lower stratosphere. These values can be compared to observations of stratospheric $Br_y$ calculated with photochemical modelling using balloon-borne direct sun DOAS (Differential Optical Absorption Spectroscopy) BrO observations (Dorf et al., 2006; Carpenter et al., 2014)



and annual mean mixing ratios derived from ground-based UV-visible measurements of stratospheric BrO (Sinnhuber et al., 2002; Hendrick et al., 2007; Hendrick et al., 2008;

Carpenter et al., 2014). These observations show the temporal development of $Br_y$ in dependence of the year when air masses are entering the stratosphere. Assuming a mean age of air of 6 years at 25 km (Haenel et al., 2015) we can compare the measured $Br_y$ from MIPAS-B directly to the $Br_y$ from DOAS observations in the years (of stratospheric entry) 2005 and 2008. In these years, the range of expected $Br_y$ spans from about 18 to 25 pptv taking into account the

error limits. Although the amount of $Br_y$ inferred from MIPAS-B measurements lies more towards the upper edge of this range, it is still consistent with the $Br_y$ estimates from DOAS observations.

Finally, it should be mentioned that there is still some limited potential on the improvement of the spectroscopy of the interfering species (mainly $HO_2NO_2$) in the $BrONO_2$ spectral analysis

window (Wagner and Birk, 2016). However, $BrONO_2$ test retrieval simulations for MIPAS-B (within this work) and MIPAS (Höpfner et al., 2009) have shown that future improvements in the spectroscopic database will most probably not exceed the total error limits given in this study.

**Acknowledgements**

We are grateful to the CNES balloon team for excellent balloon operations and the Swedish Space Corporation for operating Arctic campaigns and logistical assistance. We thank Katja Grunow from Free University of Berlin for meteorological support. The work presented here was funded in part by the European Space Agency (ESA) and the German Aerospace Center

(DLR). We acknowledge support by Deutsche Forschungsgemeinschaft and Open Access Publishing Fund of Karlsruhe Institute of Technology.



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




**Table 1.** Overview of MIPAS balloon flights and number of limb sequences recorded around sunrise (Kiruna) and sunset (Timmins). Measurement times are given in UTC and local solar time (LST) together with the solar zenith angle (SZA). Latitude and Longitude refer to the tangent points of the observations.

| Location | Date | UTC | LST | SZA (deg) | # Seq. | Latitude (°N) | Longitude (°E) |
|---|---|---|---|---|---|---|---|
| Kiruna | 24 Jan 2010 | 06:17 – 10:21 | 08:13 – 12:36 | 98.1 – 86.2 | 19 | 69.3 – 66.9 | 28.8 – 33.7 |
| Kiruna | 31 Mar 2011 | 02:00 – 04:38 | 04:01 – 06:34 | 99.4 – 83.1 | 12 | 64.0 – 63.5 | 30.1 – 28.9 |
| Timmins | 7/8 Sep 2014 | 21:40 – 02:33 | 16:25 – 21:00 | 69.9 – 115.1 | 37 | 45.9 – 46.2 | -78.8 – -83.2 |






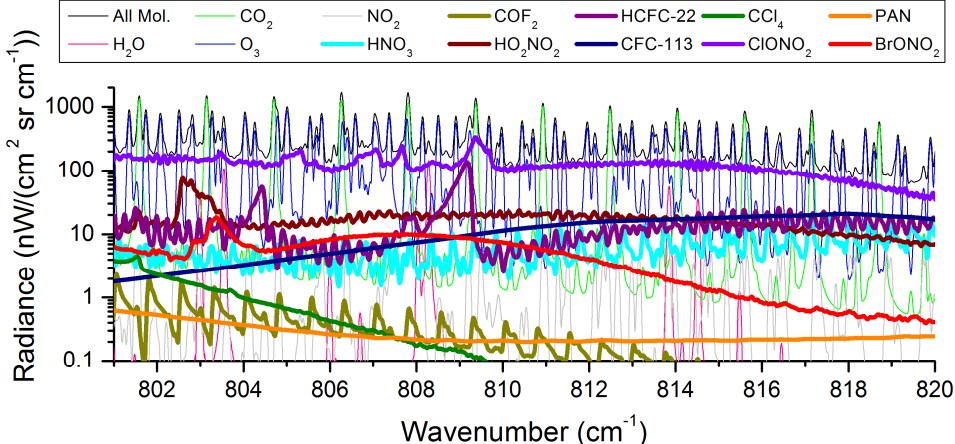

**Figure 1.** Simulated limb emission spectra (with spectral resolution of MIPAS-B) for a mid-latitude summer standard atmosphere (Remedios et al., 2007) in the spectral region of the $BrONO_2$ analysis window for a tangent altitude of 20 km. Emissions of individual species contributing to the combined spectrum (all molecules, black line) are shown.



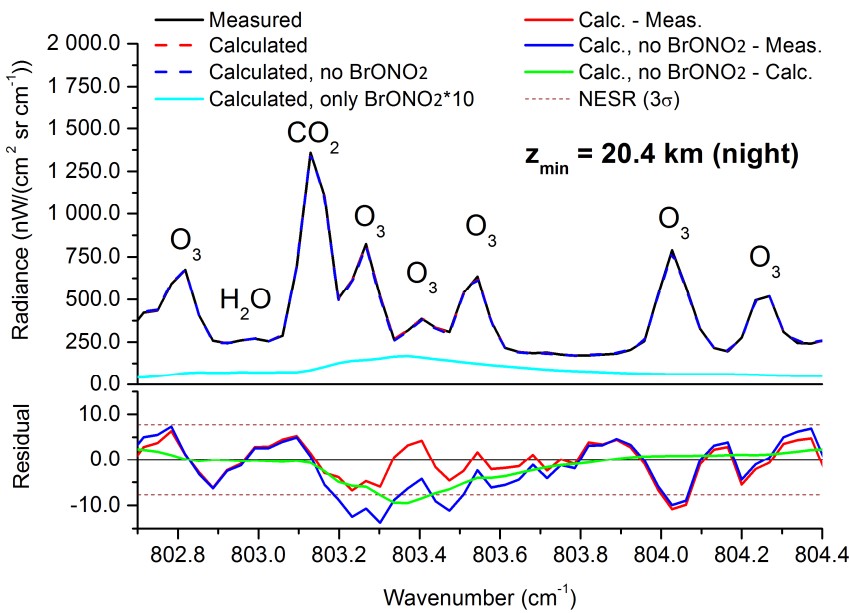

**Figure 2.** Top panel: Best fit of measured spectrum (black solid line) zoomed around the Q-branch of the $BrONO_2$ $v_3$ fundamental band at 803.37 cm$^{-1}$ for a tangent altitude ($z_{min}$) near 20 km recorded during night on 7/8 September 2014 above Timmins (Seq. 05a). A calculation with (red dashed line) and without (blue dashed line) $BrONO_2$ in the model atmosphere was performed. The calculated individual emission of the $BrONO_2$ band (scaled by a factor of 10; cyan solid line) is shown, too. Bottom panel: Difference between the calculated and measured spectrum (red solid line); difference between the calculated spectrum (without $BrONO_2$) and the measured one (blue solid line); difference of both calculations (green solid line). The 3-σ NESR (brown dotted line) is displayed, too.





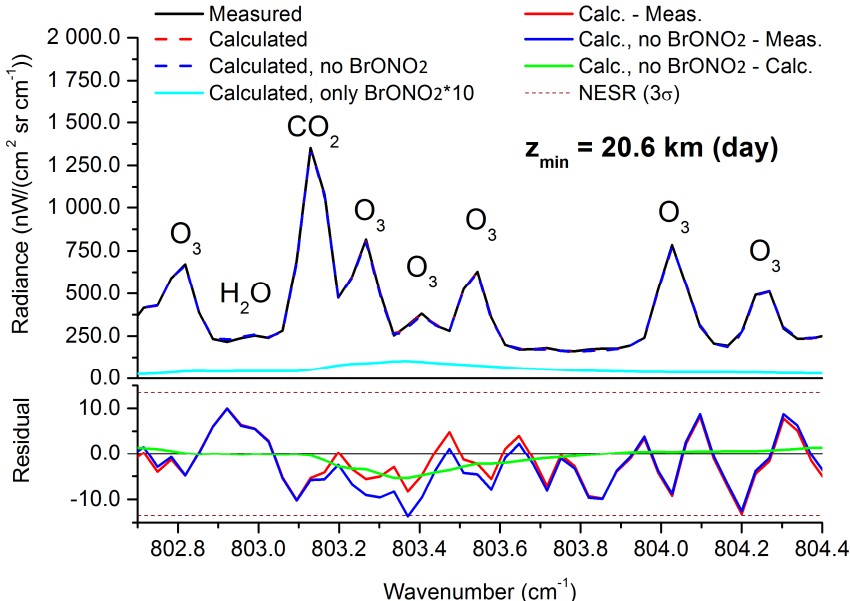

**Figure 3.** Same as Figure 2 but for a spectrum observed during day (Seq. 02e). The difference between the red and blue solid line (bottom panel) is smaller than the corresponding nighttime difference shown in Figure 2. Hence, $BrONO_2$ amounts seen during day are lower than the ones observed at night.



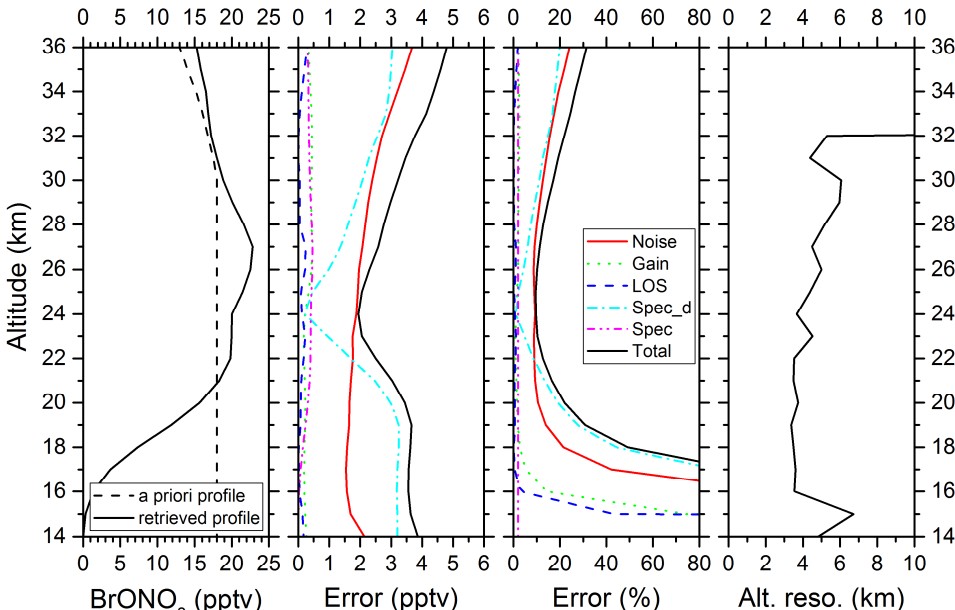

**Figure 4.** Retrieved BrONO₂ VMR vertical profile (and a priori profile) for a nighttime (Seq. 05a) limb sequence recorded by MIPAS-B on 7/8 September 2014 above Timmins together with absolute and relative errors and the altitude resolution, determined from the full width at half maximum of the columns of the averaging kernel matrix. The following error contributions are shown: spectral noise (red solid line), radiometric gain (green dotted line), LOS (blue dashed line), spectroscopic data of disturbing gases (dash dotted cyan line), spectroscopic data of target molecule BrONO₂ (short dash dotted magenta line), and total error (black solid line).




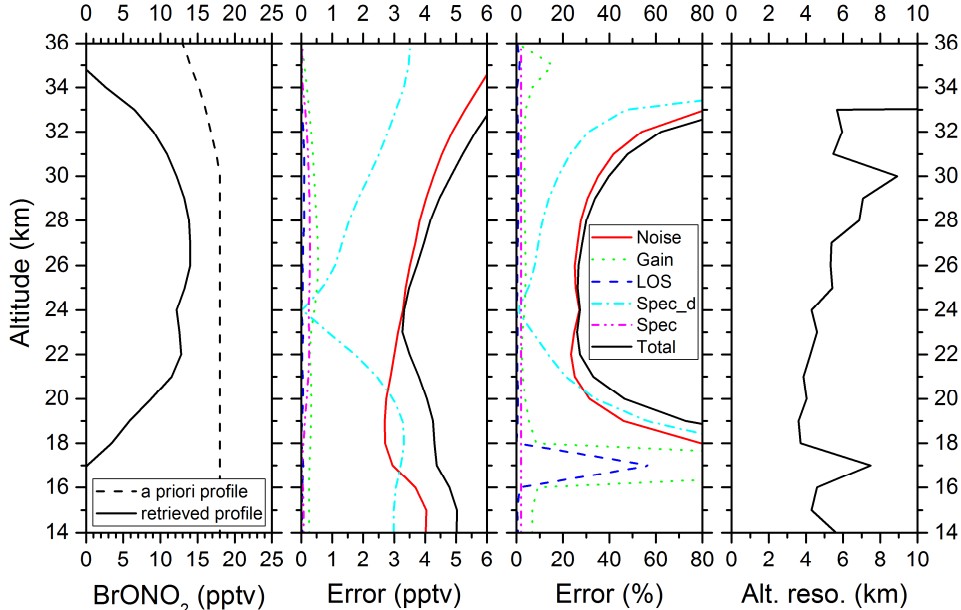

**Figure 5.** Same as Figure 4 but for a limb sequence measured during day (Seq. 02e).





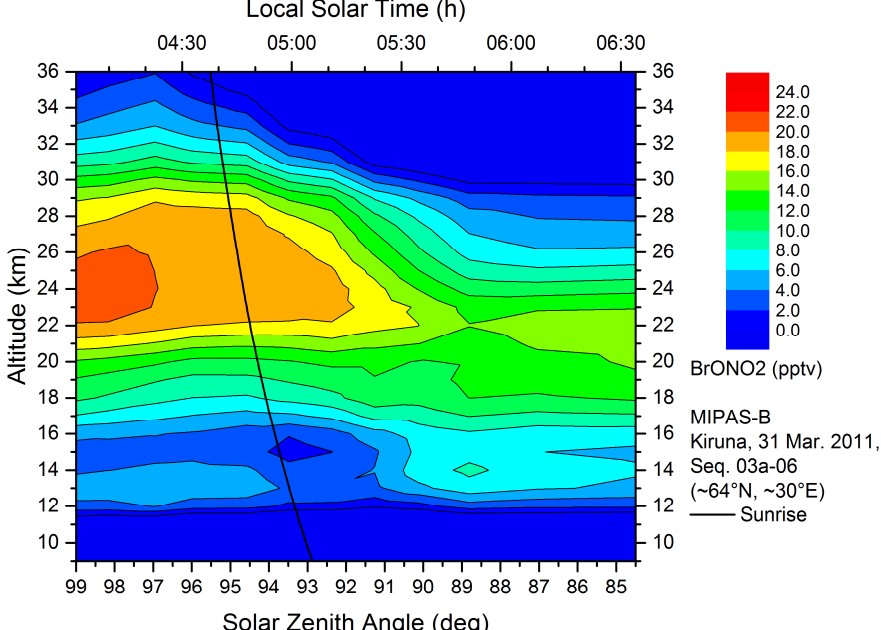

**Figure 6.** Temporal evolution of BrONO$_2$ volume mixing ratios (pptv) as seen by MIPAS-B from a float altitude around 35 km above northern Scandinavia on 31 March 2011 inside the late winter Arctic vortex. The black solid line marks the sunrise terminator. A decrease in the BrONO$_2$ amount starting around sunrise is clearly visible.





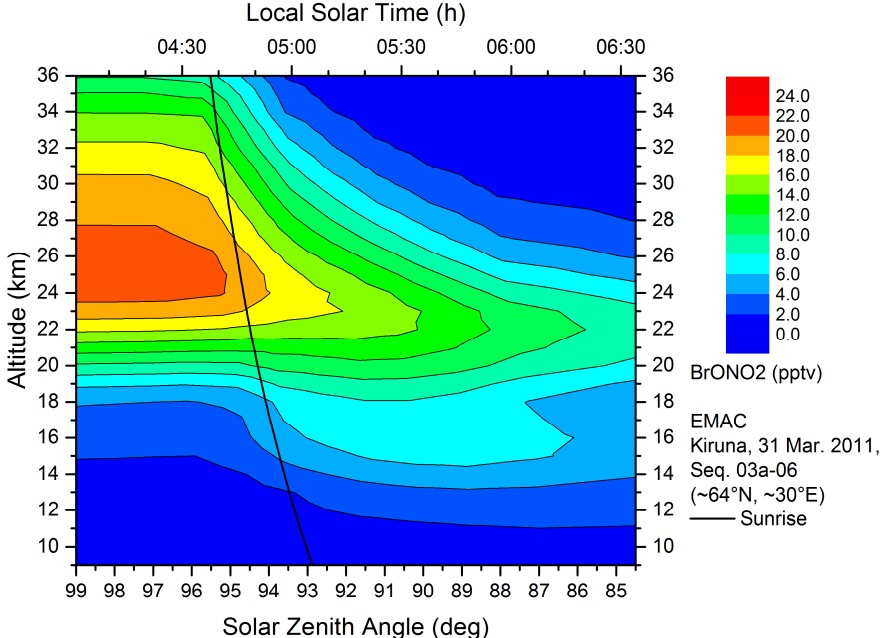

**Figure 7.** Temporal evolution of BrONO$_2$ on 31 March 2011 as simulated by the chemistry climate model EMAC. The decrease of BrONO$_2$ starts close to sunrise.



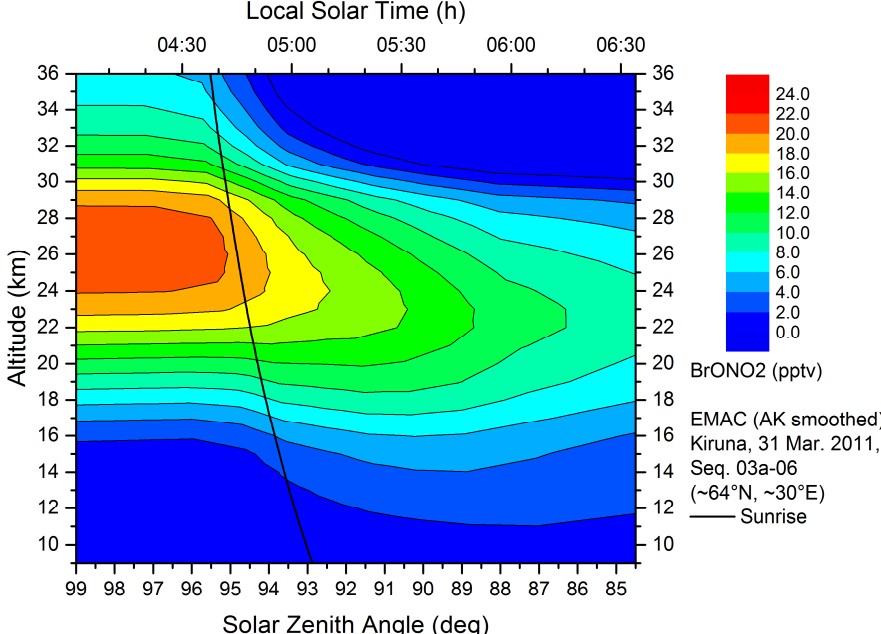

**Figure 8.** Same as Figure 7 but EMAC vertical profiles smoothed with the MIPAS-B averaging kernel (AK).





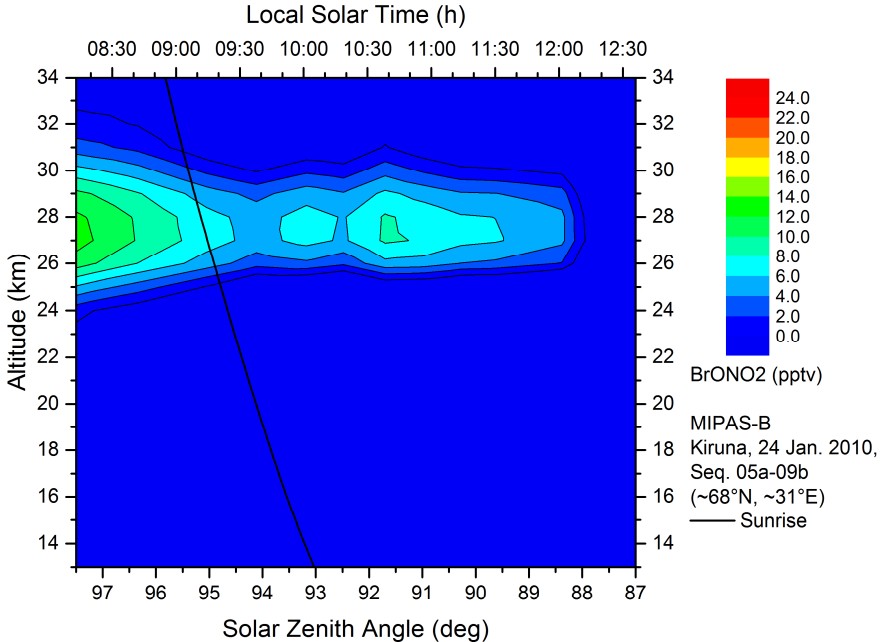

**Figure 9.** Temporal evolution of BrONO$_2$ volume mixing ratios (pptv) as measured by MIPAS-B on 24 January 2010 inside the mid-winter Arctic vortex (observer altitude about 34 km). The black solid line marks the sunrise terminator. The still weak illumination at the end of the polar night is responsible for the small diurnal variation of the BrONO$_2$ amount.






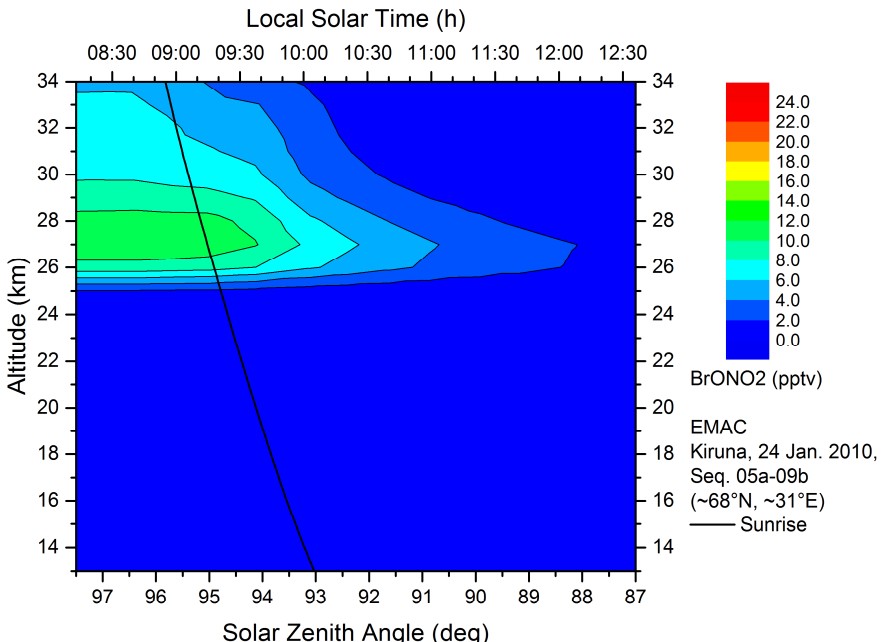

**Figure 10.** Temporal evolution of $BrONO_2$ on 24 January 2010 as simulated by the chemistry climate model EMAC. The decrease of $BrONO_2$ starts close to sunrise.





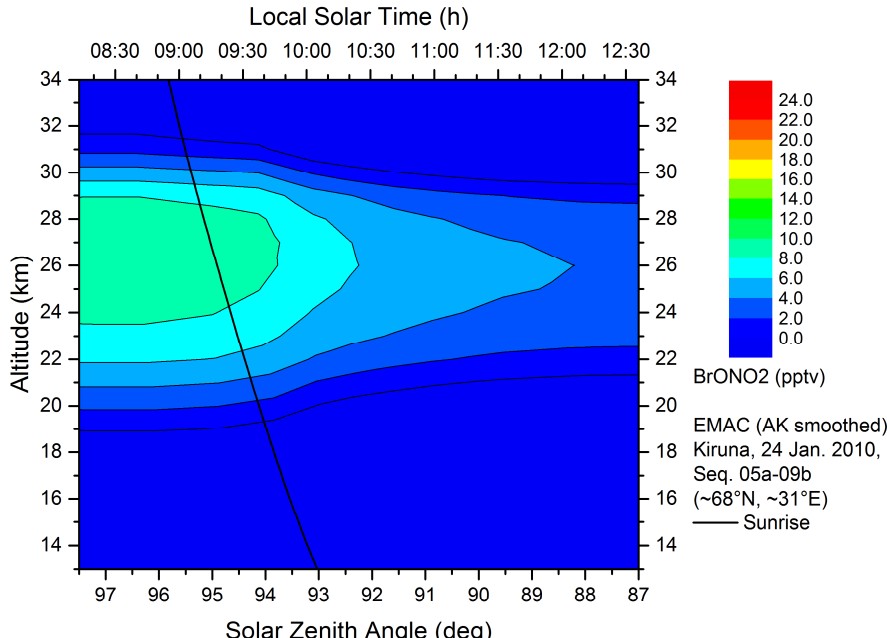

**Figure 11.** Same as Figure 10 but EMAC vertical profiles smoothed with the MIPAS-B averaging kernel (AK).



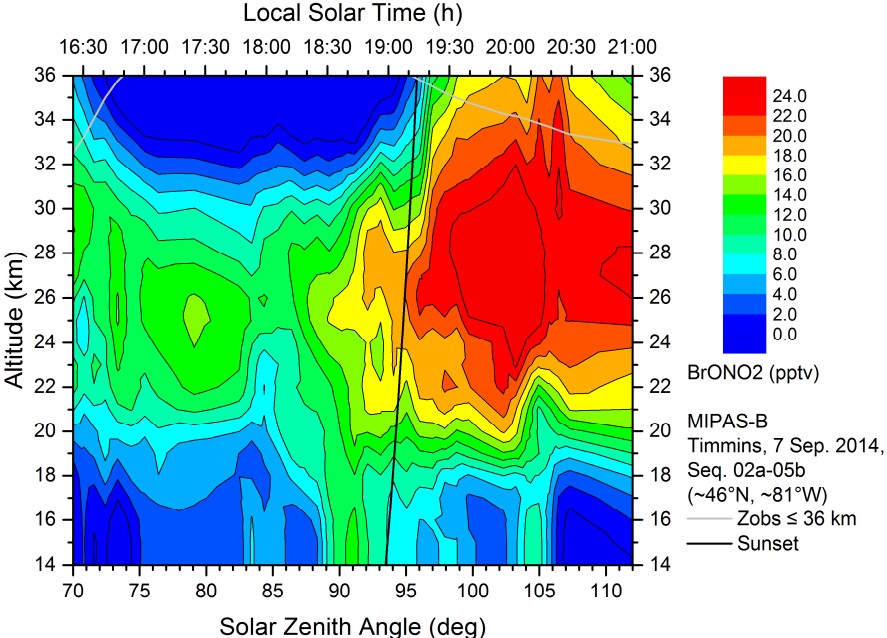

**Figure 12.** Temporal evolution of BrONO$_2$ amounts observed by MIPAS-B near 46°N above Ontario (Canada) on 7 September 2014. The grey line indicates the time periods where the balloon gondola float altitude was lower or equal to 36 km. The black solid line marks the sunset terminator. The build-up of BrONO$_2$ from daytime BrO starts shortly before sunset.





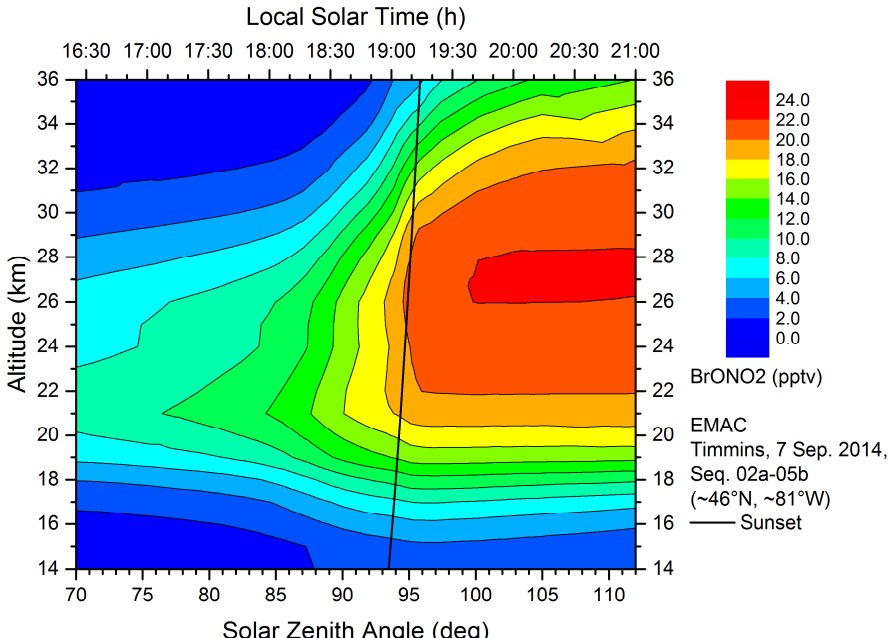

**Figure 13.** Temporal evolution of BrONO$_2$ on 7 September 2014 as simulated by the EMAC model.



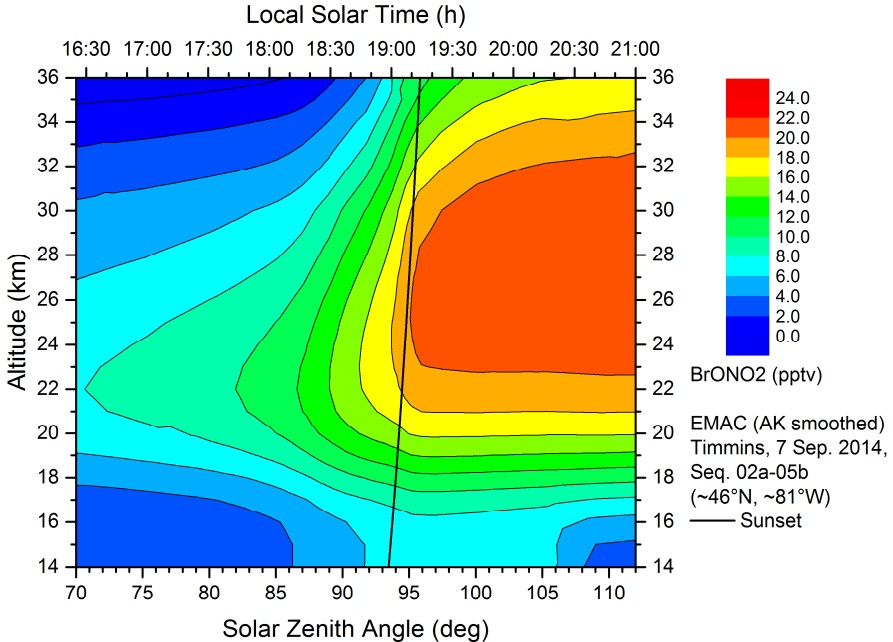

**Figure 14.** Same as Figure 13 but EMAC vertical profiles smoothed with the MIPAS-B averaging kernel (AK).
