# Peer review of "Diurnal variations of BrONO2 observed by MIPAS-B at mid-latitudes and in the Arctic"

_Atmospheric Chemistry and Physics, 2017_

## Referee Comment (RC1) · Anonymous Referee #1 · 14 Aug 2017

General comments:

This paper discuss temporal variations of BrONO$_2$, the nighttime main reservoir of Bry, in the polar and mid-latitude stratospheres. They clearly showed different behaviors of BrONO$_2$ in nighttime: the amounts of BrONO$_2$ revealed more than 20 pptv in the absence of PSCs and less than 14 pptv in the presence of PSCs. These values were well reproduced by CTM for which the averaging kernel matrix of MIPAS-B was applied. Thus, temporal behaviors of BrONO$_2$ during sunset/sunrise are thought to be understood by conventional chemical reactions involving radicals such as ClOx etc. With this knowledge, they estimated the amount of Bry to be 21-25 pptv in the lower stratosphere. I would suggest that this paper should be published in ACP; however, some messages to the reader would be desired or clarified before the acceptance of

this manuscript.

1. Which is better for estimating Bry: daytime BrO or nighttime $BrONO_2$?

I would suggest to add some discussion about the estimate of Bry. If we select a condition where no heterogeneous reactions occur, is the measurement of $BrONO_2$ in nighttime a better way? The authors state comparison with previous studies in "Conclusion" without any discussion about it in "Results and discussion". Thus, I suggest to add a subsection, e.g., "Comparison with other studies", then discuss about studies on BrO measurements, the estimation of Bry, and the advantage of $BrONO_2$ in the estimation, if so.

2. Are heterogeneous reactions on sulfate not important for the destruction of $BrONO_2$ in nighttime under volcanically quiescent periods and temperatures observed?

Under conditions where no PSCs were evident in the Arctic March and the mid-latitude September, significant enhancements of $BrONO_2$ up to 21-22 pptv were measured by MIPAS-B. This may suggest that any heterogeneous reactions (or hydrolysis) of $BrONO_2$ on sulfate is not important, at least, under such a low aerosol surface area density and temperatures.

Minor points:

1. page 5, line 130: What is instrumental offset? The authors mention that continuum could be separated from individual spectral lines.

2. page 6, Figure 6: What is a cause of difference in peak altitudes? Namely, 24 km in nighttime and 22 km in daytime. This feature is also seen from the model result, so that the authors can provide some explanation for that. In connection with this, additional figures from model computations are useful, if the authors provide figures showing difference in the partitioning of Bry species at day and night with and without PSCs. Then, add some discussion on that.

3. page 7, line 209: is it right for this calculation, because the model grid $(x - x_a{}^*)$ is

larger than that of MIPAS-B ($x_a$)?

4. page 9, line 274: The authors state "starts earlier". What is the difference in time? I suggest to write: e.g., "The BrONO$_2$ increase stats at XXXXUT in the measurement, whereas the model BrONO$_2$ increase starts at YYYYUT."

---

## Referee Comment (RC2) · Anonymous Referee #2 · 31 Aug 2017

The paper of Wetzel et al. presents the first stratospheric measurements of BrONO2 at mid- and polar latitudes under twilight conditions. These measurements were carried out using the balloon-borne MIPAS-B instrument launched from Kiruna (68°N, Sweden) in January 2010 and March 2011, and from Timmins (49°N, Canada) in September 2014. Observed BrONO2 vertical profile temporal variations are compared to multi-year simulations performed by the EMAC CTM and the latter is found to capture well the 14 and 20 pptv of BrONO2 measured in the 22-25 km altitude range in the presence and absence of PSCs, respectively. Total inorganic bromine (Bry) amount is then derived by combining MIPAS-B BrONO2 concentrations with modelled BrONO2/Bry ratios. The estimated Bry values are about 21-25 pptv in the lower stratosphere, i.e. at the upper limit of the already published estimates calculated from BrO DOAS observations.

The manuscript is well written and clearly structured. I recommend its publication in ACP after addressing the following comments:

General comments:

1/The comparisons between measured and modelled BrONO2 mixing ratios are discussed in a too qualitative way and therefore it is very difficult for the reader to have a quantitative view about the level of agreement (or discrepancy) between MIPAS-B and EMAC. To improve this, I suggest to add in the manuscript 2D colorplots of EMAC minus MIPAS-B BrONO2 VMR relative differences for the three balloon flights and for both smoothed and unsmoothed model profiles. Those plots would also help to better characterize and discuss the impact of the smoothing of model profiles by MIPAS-B averaging kernels on the comparison results (see e.g. page 9, lines 255-261).

2/As Anonymous Referee #1, I strongly recommend to discuss the pros and cons of using nighttime BrONO2 for estimating Bry, instead of daytime BrO.

Specific comments:

1/Page 6, lines 169-171: The authors should briefly described here how the vertical resolution of the MIPAS-B BrONO2 observations is estimated (FWHM of the averaging kernel matrix). I think that showing typical averaging kernels could be also useful to see in which altitude range the maximum sensitivity of the MIPAS-B BrONO2 measurements is located.

2/Page 8, lines 223-227: It is stated that the sensitivity study of Kreycy et al. (2013) about the BrONO2 photolysis rate is not relevant here because it was conducted for mid-latitude conditions. If this is true for the two Kiruna flights, this is not the case for the third flight, which was launched from Timmins (49°N, Canada), i.e. at mid-latitude. For the latter, I would suggest to also perform model simulations using the Keycy et al. BrONO2 photolysis rate and see how it impacts (1) the MIPAS-B versus model

comparison, and (2) the Bry estimate.

---

## Author Comment (AC1) · 20 Sep 2017

**Response to Referee #1:**

First of all we thank the referee for the effort to carefully reading the manuscript and for all comments.

**General comments:**

*1. Which is better for estimating Bry: daytime BrO or nighttime BrONO2? I would suggest to add some discussion about the estimate of Bry. If we select a condition where no heterogeneous reactions occur, is the measurement of BrONO2 in nighttime a better way? The authors state comparison with previous studies in "Conclusion" without any discussion about it in "Results and discussion". Thus, I suggest to add a*

[Figure]

*subsection, e.g., "Comparison with other studies", then discuss about studies on BrO measurements, the estimation of Bry, and the advantage of BrONO2 in the estimation, if so.*

First, we have to emphasize that MIPAS-B can only measure BrONO2 but not BrO such that we are only able to estimate Bry from BrONO2 data. However, we performed chemical model calculations with EMAC and a 1-D stacked box model to look at the diurnal variation of BrONO2 and BrO. As already mentioned in the text in section 3.2, more than 90% of Bry are in the form of BrONO2 between 21 and 29 km during the night at mid-latitudes in September. In contrast, the maximum relative daytime amount of BrO is only 80% of Bry near 36 km, the upper altitude limit of this MIPAS-B measurement. Furthermore, the BrO VMR is gradually changing during day (while BrONO2 VMR is rather constant during night). Hence, BrONO2 appears to be better suited to estimate Bry compared to BrO. This holds also for the situation during the Arctic flight in March where the nighttime BrONO2/Bry ratio is larger than the daytime BrO/Bry ratio as well. We added a new Section 3.3 where we discuss the estimation of Bry together with the comparison to DOAS results.

*2. Are heterogeneous reactions on sulphate not important for the destruction of BrONO2 in nighttime under volcanically quiescent periods and temperatures observed? Under conditions where no PSCs were evident in the Arctic March and the mid-latitude September, significant enhancements of BrONO2 up to 21-22 pptv were measured by MIPAS B. This may suggest that any heterogeneous reactions (or hydrolysis) of BrONO2 on sulphate is not important, at least, under such a low aerosol surface area density and temperatures.*

Aerosol surface areas are very low in the altitude regions of the BrONO2 VMR maxima (2.0E-09 cm2/cm3 at 24 km on 31 March 2011 and 2.0E-09 cm2/cm3 at 28 km on 7 September 2014 in the EMAC simulation) and are not important in these two cases.

**Minor comments:**

*1. Page 5, line 130: What is instrumental offset? The authors mention that continuum could be separated from individual spectral lines.*

Instrumental offset is an additive radiometric parameter which is not completely eliminated during the calibration process (for all instrumental issues, see Friedl-Vallon et al., 2004 as cited in the text). We included "radiometric" in the text for better clarity.

*2. Page 6, Figure 6: What is a cause of difference in peak altitudes? Namely, 24 km in nighttime and 22 km in daytime. This feature is also seen from the model result, so that the authors can provide some explanation for that. In connection with this, additional figures from model computations are useful, if the authors provide figures showing difference in the partitioning of Bry species at day and night with and without PSCs. Then, add some discussion on that.*

The displacement of the nighttime $BrONO_2$ VMR maximum from 24 km down to 22 km during day can be explained by photolysis. Towards higher altitudes, the decomposition of $BrONO_2$ according to (R2a), (R2b), and (R3) is increasingly faster than the $BrONO_2$ build-up via (R1). We explain this in the text now and included a Figure showing BrO from the model. Further Figures showing the complete model bromine partitioning are not necessary to explain this displacement.

*3. Page 7, line 209: is it right for this calculation, because the model grid (x − xa\*) is larger than that of MIPAS-B (xa)?*

The formula is correct. The Matrix A has not a quadratic but a rectangular form to account for the different altitude grids.

*4. Page 9, line 274: The authors state "starts earlier". What is the difference in time? I suggest to write: e.g., "The BrONO2 increase starts at XXXXUT in the measurement, whereas the model BrONO2 increase starts at YYYYUT."*

It is difficult, to give exact times for the beginning of the $BrONO_2$ increases. However, we changed the corresponding text to characterize both increases more precise.

---

## Author Comment (AC2) · 20 Sep 2017

**Response to Referee #2:**

First of all we thank the referee for the effort to carefully reading the manuscript and for all comments.

**General comments:**

*1. The comparisons between measured and modelled BrONO2 mixing ratios are discussed in a too qualitative way and therefore it is very difficult for the reader to have a quantitative view about the level of agreement (or discrepancy) between MIPAS-B and EMAC. To improve this, I suggest to add in the manuscript 2D colorplots of EMAC minus MIPAS-B BrONO2 VMR relative differences for the three balloon flights and for*

*both smoothed and unsmoothed model profiles. Those plots would also help to better characterize and discuss the impact of the smoothing of model profiles by MIPAS-B averaging kernels on the comparison results (see e.g. page 9, lines 255-261).*

As proposed by the referee we added the corresponding 2D plots to the manuscript and modified the text accordingly.

*2. As Anonymous Referee 1, I strongly recommend to discuss the pros and cons of using nighttime BrONO2 for estimating Bry, instead of daytime BrO.*

We included a new section 3.3 where we discuss all the issues concerning the estimation of Bry.

**Specific comments:**

*1. Page 6, lines 169-171: The authors should briefly describe here how the vertical resolution of the MIPAS-B BrONO2 observations is estimated (FWHM of the averaging kernel matrix). I think that showing typical averaging kernels could be also useful to see in which altitude range the maximum sensitivity of the MIPAS-B BrONO2 measurements is located.*

The altitude resolution was calculated from the full width at half maximum (FWHM) of the rows of the averaging kernel matrix. This is written in the text now. The altitude resolution is shown in the right column of Figures 4 and 5. The maximum sensitivity is located in the altitude region near the measured tangent altitudes (as typical for limb sounding).

*2. Page 8, lines 223-227: It is stated that the sensitivity study of Kreycy et al. (2013) about the BrONO2 photolysis rate is not relevant here because it was conducted for mid-latitude conditions. If this is true for the two Kiruna flights, this is not the case for the third flight, which was launched from Timmins (49° N, Canada), i.e. at mid-latitude. For the latter, I would suggest to also perform model simulations using the Kreycy et al. BrONO2 photolysis rate and see how it impacts (1) the MIPAS-B versus model*

*comparison, and (2) the Bry estimate.*

There was a mistake in the manuscript. The Kreycy et al. (2013) flight was performed at the beginning of September 2009 not at mid-latitudes but from Kiruna (Sweden). Anyhow, we tested the 1.7*J/k ratio recommended by Kreycy et al. by using a 1-D photochemical stacked box model for the situation of our September 2014 flight. The photolysis rate of BrONO2 will then be enhanced and the production rate of BrONO2 will be reduced. As expected, this leads to lower BrONO2 during day towards higher BrO amounts. Hence, the Kreycy et al. recommendation further degrades the agreement between model simulations and our measurements since daytime BrONO2 values are already lower in the model (compared to the measurement) using the standard JPL kinetics. Thus, in our case, we should rather scale the J/k ratio with a factor < 1 (and not > 1 as stated by Kreycy et al.) to get a better daytime agreement between measurement and simulation. Hence, concerning the EMAC calculations, we stick on the JPL data. It is important to emphasize, that during night, the BrONO2 VMR does not change significantly (< 0.1 pptv) when varying the J/k ratio in the altitude region where the BrONO2/Bry ratio is high (> 0.90) such that our estimation of Bry from measured nighttime BrONO2 is not influenced by the outcome of the Kreycy et al. study. These conclusions are also valid for the situation during the March 2011 flight. We added some sentences in Section 3.2 to explain these issues in the manuscript.